# The Effect of Sense of Community Responsibility on Residents’ Altruistic Behavior: Evidence from the Dictator Game

**DOI:** 10.3390/ijerph17020460

**Published:** 2020-01-10

**Authors:** Chao Yang, Yanli Wang, Yuhui Wang, Xuemeng Zhang, Yong Liu, Hong Chen

**Affiliations:** 1Faculty of Psychology, Southwest University, Chongqing 400715, China; yangchaopsy632@163.com (C.Y.); wyl1054616437@163.com (Y.W.); zxuemengzfb@163.com (X.Z.); psyliuy@email.swu.edu.cn (Y.L.); 2Key Laboratory of Cognition and Personality (Ministry of Education), Southwest University, Chongqing 400715, China; 3School of Psychology, Guizhou Normal University, Guiyang 550025, China; 4Department of Psychology, Renmin University of China, Beijing 100872, China; yuhui2016@ruc.edu.cn

**Keywords:** sense of community responsibility, altruistic behavior, dictator game, social norms

## Abstract

Understanding the new mechanism of altruistic behavior is pivotal to people’s health and social development. Despite the rich literature on altruism, this is the first study exploring the association between the sense of community responsibility (SOC-R) and altruistic behavior by repeated dictator games. Data were gathered from 95 residents (30% male; *M* age = 33.20 years). Demographic variables, money motivation, and SOC-R were measured. The results revealed that there was a significant positive correlation between SOC-R and altruistic behavior, and SOC-R had a positive predictive effect on residents’ altruistic behavior. With the increasing of the number of tasks assigned, the level of residents’ altruistic behavior gradually decreased. There was a significant difference in money allocation between the groups with high and low levels of SOC-R. The level of altruistic behavior in the group with a high level of SOC-R was significantly higher than that in the the group with a low level of SOC-R. Findings from the present study highlighted the potential value of strengthening residents’ SOC-R in the improvement of altruism. Implications and directions for future research were also discussed.

## 1. Introduction

Many meta-analytic, horizontal and vertical studies have shown that people’s concern for others has begun to decrease since 1979 and reached its largest decline between 2000 and 2010 [1]. Meanwhile, researchers have found that people’s civic values (such as concern about social issues and environmental protection behaviors) also declined [2]. Moreover, the 2013 China General Social Survey (CGSS) showed that only 32% of 5734 residents joined in donations voluntarily and unconditionally, and only 7% of the residents participated in social volunteer service activities (the survey can be accessed online at http://www.cnsda.org/index.php?r=projects/view&id=93281139). In 2017, the Chinese government proposed the abandonment of egoism and stated that altruism is an effective way to build a community of human destiny. In summary, with the development of society, people’s citizenship cognition and behavior may be out of sync with the rapid growth of material consumption, especially for people living in developing countries. Making individuals feel responsible and care for society and others should be a topic of common concern for current researchers and managers.

Altruistic behavior is a special type of social behavior, which is defined as sacrificing one’s resources to benefit others without expecting external rewards [3], such as blood donation and giving instrumental help. It not only guarantees people’s mental and physical health [4,5] but also has important significance for human collective corporation and social development [6]. Previous studies have mainly explained the mechanism of altruistic behavior from an evolutionary–theoretical perspective. It was thought there were two types of altruism: Kin altruism [7] and reciprocal altruism [8]. This means that people tend to help relatives and persons with whom they exchange benefits, to ensure that family genes are passed on to future generations and that their psychology is kept in balance between giving and receiving. Besides, it was argued that empathy [9] and a sense of guilt [10] also affected individual’s altruistic behavior. The majority of research conducted on altruistic behavior has used the Self-Report Altruism scale (SRA scale) [11] and dictator game (DG) [12]. The SRA scale is a unidimensional scale comprising 20 items describing altruistic acts in various scenarios, and responses are given on a five-point Likert scale ranging from “never” to “always”. However, some items of SRA scale are not suitable for community residents, and some scenarios are not common in China; for example, “I have helped a classmate who I did not know very well with a homework assignment because my knowledge of the topic was greater than his/hers”. Therefore, we chose the DG method in this study. 

Previous studies have suggested that DG is an effective paradigm to evaluate an individual’s altruistic behavior [13,14]. It is an economic game in which two players are unknown to each other. The standard paradigm is a one-shot task [12]. At the beginning of the game, the participants will be divided into dictators and receivers. Meanwhile, the dictator will receive a certain amount of money from the experimenter, and he can dispense any amount of money to the receiver. Whatever the amount is, the receiver can only accept and has no power to reject the proposal. The more money the dictator allocates to the receiver, the more altruistic he is. A meta-analysis finds that the dictator usually allocates 28.35% of the amount to the receiver [15]. Comparing with the ultimatum game, in which receiver can reject the proposal, DG eliminates the dictator’s fear of the receiver’s rejection and strategy processing based on reciprocity motivation [16]. Besides, in comparison with real donor behavior, DG can more accurately reflect the individual’s altruistic behavior because of its anonymity and its direct involvement with the individual’s interests. However, real donor behavior is susceptible to the level of individual empathy [17]. 

However, researchers have found that the level of individual altruism changes dynamically. Scholars found that the dictator allocated less money to the receiver when there were more receivers [18], which meant people would become selfish in repeated dictator games (DGs). Researchers thought it was not possible to determine one’s level of altruism just relying on the amount of money he/she gave in the one-shot DG [16]. Thus, researchers developed the repeated DGs, in which dictators needed to complete multiple rounds of assignments with a different partner for each trial [16,19]. Meanwhile, previous research usually used college students as subjects [18]. However, a meta-analysis found that non-student participants gave much more money to the receiver than student participants [15]. Therefore, the related conclusions of students may not apply to non-students (e.g., residents) and the repeated DGs are more suitable for assessing an individual’s altruistic behavior.

A wide range of research has been done on the SOC-R [20,21,22]. It refers to a feeling of selfless personal responsibility for both the individual and the collective well-being of a community [20,23]. It originates from scholarly discussion of the sense of community (SOC) [20,23]. It is connected but different from SOC. Specifically, it is dominated by personal information systems, which reflects the sense of responsibility of residents to the community and others. On the contrary, SOC is based on the theory of needs, emphasizing the importance of the community to meet people’s physical and psychological needs. SOC-R improves the research on SOC and deepens it [21]. It is a very important concepts in community psychology. SOC-R is usually measured by the Sense of Community Responsibility Scale (SOC-RS), which is a unidimensional scale with six items [20]. Then, Treitler et al. [21] added two items to the scale so that it could have higher discriminant validity. However, most of the existing research has only focused on the effect of SOC-R on employees’ well-being, work satisfaction, and organizational citizenship behavior [24,25]. There is little research on resident SOC-R out of organizational context, and the items of SOC-RS are macroscopic. Moreover, the operational definition of SOC-R is vague. Nowell and Boyd [20] suggested that future studies should advance and apply the theory of SOC-R in a wide array of settings, because the items of SOC-R were specially designed for the community collaborative context. Hence, with qualitative and quantitative research, Yang, Chen, Hu, and Wang [22] developed the Sense of Community Responsibility Questionnaire (SOC-RQ) for Chinese residents, and it had good reliability and validity. They defined SOC-R as a kind of attitude including the cognition of self-responsibility to other people and relationships in the community, the willingness to emotionally invest in the well-being and collective interests of others, and the tendency to put words into action and accept the consequence of actions. By contrast, Yang et al. [22] gave a more detailed psychological definition of SOC-R; their definition is consistent with the theory of attitude [26], and the items of SOC-RQ are taken from residents’ daily life, which can perhaps better reflect the level of residents’ SOC-R. Therefore, we adopted this definition and used SOC-RQ to measure the SOC-R in the present study. 

At present, the direct relationship between SOC-R and altruistic behavior is not known. Hoffman [27] thought altruism was a sense of responsibility for others and the manifestation of the development of individual responsibility. Brebels, De Cremer, and Sedikides [28] deemed that the perception of responsibility was the basis of altruism and the activation of social norms. Moreover, research on children’s altruism showed that the responsible group expressed more altruism than the non-responsible group [29]. A study demonstrated that social responsibility was a determinant of altruistic behavior [30]. Individuals with a stronger sense of family responsibility reported more altruistic behavior [31]. These studies indicate that there is a relationship between responsibility and altruistic behavior. According to the theory of SOC-R [23], personal belief systems make people’s behavior conform to social norms. The core of this theory is that social norms make people’s cognition and behavior coordinate and then adapt to normative requirements of different social situations. The early theory of social norms believed that individual’s altruistic behavior was influenced by people’s perception of responsibility of social norms [32]. Social norms make people altruistic in the DG [18,33]. Moreover, a study indicated that the sense of responsibility had cross-context consistency: An individual’s sense of responsibility in one context (e.g., family) had a strong predictive effect on his sense of responsibility in another context [34]. People not only have a need to be a part of community but also feel a sense of duty and responsibility toward their community [35,36]. Therefore, perhaps we can speculate on the relationship between SOC-R and altruistic behavior from studies of family responsibility and social responsibility. 

In the past, researchers paid little attention to the altruistic behavior and SOC-R of residents (especially Chinese residents). Despite rich literature on altruism, this is the first study exploring the association between SOC-R and altruistic behavior by repeated DGs. Based on the theory of SOC-R and social norms as well as the relationship between responsibility and altruism, the present study aimed to verify the relationship between SOC-R and altruistic behavior to provide a theoretical reference for the cultivation of residents’ altruistic behavior. 

## 2. Methods

### 2.1. Participants

Participants (N = 95, 30% male) were urban residents from Southwest China. They came from many different communities and were recruited through community posters. Participants ranged between 18 and 50 years old (*M* = 33.20, *SD* = 9.22). Most of them were Han ethnicity (95.79%), reported no religious beliefs (88.42%), owned their homes (92.63%), and lived in their current communities for more than 10 years (66.30%). Their vision or corrected vision was normal, and they had no color blindness or weak color and were familiar with the basic operation of a computer.

### 2.2. Measures

The SOC-RQ [22] questionnaire consists of 22 items across three dimensions including community responsibility cognition, community responsibility emotion, and community responsibility behavior. Examples of items are “I think that residents should not do anything that is harmful to the community”, “I will be proud to participate in the community representative election” and “I will take part in community service activities.” Items were scored using a five-point Likert scale ranging from 1 = not true at all to 5 = absolutely true. For the present study, Cronbach’s alpha was 0.91.

### 2.3. Design and Procedure

This study was a single-factor experimental design. The independent variable was SOC-R, which included groups with a high level of SOC-R and low level of SOC-R, and the dependent variable was altruistic behavior, which was assessed by the average offer that the dictator assigned to the receivers. 

Referring to the paradigm of repeated DGs [16,19], each participant needed to perform three rounds of DGs. Since the task only focused on the dictator’s altruism, the three receivers were virtual. In order to balance the influence of the sequential effect, half of the participants were first given the questionnaire, and the other half first took part in the repeated DGs. Moreover, researchers found that the amount of money had no effect on the result [37,38]. Meanwhile, there was a significant positive correlation between the results with virtual money and real money [39]. Therefore, we used 10 CNY (1.49 USD) as the material for this study. The experimental procedure and materials were edited and presented by E-prime 2.0. 

The total process included four steps: (1) reading and signing the informed consent form, (2) filling in the questionnaire, (3) completing the repeated DGs, and (4) making a choice on the money motivation scale. Referring to the experimental design of Achtziger et al. [16], a seven-point Likert scale was added after DGs to assess the impact of individual money motivation on the result. The question was, “How much has the possibility of earning money motivated you to participate or perform as well as possible in this experiment?”. To minimize the communication between residents, the participants were asked to keep the contents of the experiment confidential in the informed consent form. After all participants completed the experiment, we explained the true purpose of the experiment and gave them 15 CNY (2.23 USD) for their participating. This study was approved by the university behavioral research ethics board (H17023).

The detailed experimental process was as follows (see Figure 1). First, a red gaze point was presented in the center of the screen for 500 milliseconds. Then, the ID number of a virtual receiver was shown up at pseudo-random to make the experiment more credible. In this screen, the dictator needed to make a choice from zero to ten CNY by using their mouse. A blank screen appeared for a duration of 1000 milliseconds; then, the program went to the next trial. The experiment consisted of practice and formal experiments. In order to increase the authenticity of the experiment, the practice experiment consisted of an assigning task and accepting task. The purpose was to let the participant understand the operation and believe that he (she) would probably be a dictator or receiver. Meanwhile, participants were told that their partners came from the same community. The formal experiment included three trials, and all participants were dictators.

## 3. Results

### 3.1. The Difference of Dictators’ Giving in Different Trials

Descriptive analysis showed that participants exhibited different levels of altruistic behavior in different trials (see Table 1). The minimum offer was 0 and the maximum offer was 10, which meant that people had a difference between complete self-interest and complete altruism. The overall average offer was 5.13, indicating that participants had a preference for fairness. Further, a one-sample *t*-test found that the average offer of trial 1 was significantly larger than that of trial 2 (*t* (94) = 4.03, *p* < 0.001, Cohen’s *d* = 0.41) and trial 3 (*t* (94) = 8.34, *p* < 0.001, Cohen’s *d* = 0.86). At the same time, the average offer of trial 2 was significantly larger than that of trial 3 (*t* (94) = 4.42, *p* < 0.001, Cohen’s *d* = 0.46). In conclusion, the amount of money allocated to others gradually decreased with the increasing of the round of DGs (see Figure 2).

### 3.2. The Effect of SOC-R on Altruistic Behavior

Correlation analysis showed that there was no significant correlation between money motivation and the dictators’ average offers (*r* = 0.06, *p* = 0.56). Taking average offers as the outcome variable and money motivation as the categorical variable, we tested the effect of money motivation on the altruistic behavior. First, we divided the participants into high (*M* = 6.52, *SD* = 0.51) and low (*M* = 2.19, *SD* = 0.90) motivation subgroups according to 27% of the highest and lowest scores of money motivation. Then, the independent-sample *t*-test showed that there was no significant difference between these subgroups (*t* (51) = 0.61, *p* = 0.55) (see Figure 3a). Specifically, the average offer of the high money motivation group was 5.33 (n = 27, *SD* = 1.67), and the average offer of the low money motivation group was 5.01 (n = 26, *SD* = 2.15). It could be considered that the level of the money motivation of residents did not have a significant influence on the experimental results.

Meanwhile, the correlation analysis showed that SOC-R was significantly positively correlated with altruistic behavior (*r* = 0.47, *p* < 0.001). This revealed that SOC-R was likely to affect residents’ altruistic behavior. In turn, residents’ altruistic behavior reflected the differences in residents’ level of SOC-R. Taking average offers as the outcome variable and SOC-R as the categorical variable, we divided participants into a high SOC-R group (*M* = 4.68, *SD* = 0.17) and low SOC-R group (*M* = 3.45, *SD* = 0.30) by the same grouping method. The independent-sample *t*-test showed that the average offers between these groups was significantly different (*t* (51) = 3.50, *p* < 0.01, Cohen’s *d* = 0.72) (see Figure 3b). Specifically, the average offer of the high SOC-R group was 5.94 (n = 27, *SD* = 2.06), and the average offer of the low SOC-R was 4.13 (n = 26, *SD* = 1.70). This showed that residents with different levels of SOC-R had different levels of altruistic behavior.

To further examine the impact of SOC-R on altruistic behavior, we also tested the linear relationship between them. With SOC-R as the predictor and altruistic behavior as the outcome variable, regression analysis showed that there was a significant linear relationship between residents’ SOC-R and altruistic behavior (*F* (1, 93) = 26.86, *p* < 0.001). SOC-R could explain 22.4% of the variability of altruistic behavior (*R*^2^ = 0.224). The regression equation established was y = −2.16 + 1.78x. In this equation, y represents the average level of altruistic behavior of residents and x represented SOC-R. In conclusion, SOC-R had a positive predictive effect on residents’ altruistic behavior.

## 4. Discussion

This study validated the relationship between SOC-R and altruistic behavior through DGs. The results showed that, as the number of DGs increased, the level of altruistic behavior of residents gradually decreased. The SOC-R was significantly positively correlated with residents’ altruistic behavior. Meanwhile, residents with different levels of SOC-R had different levels of altruistic behavior. First, residents with high levels of SOC-R were more altruistic than those with low levels of SOC-R. Second, SOC-R could positively predict residents’ level of altruistic behavior. In conclusion, SOC-R had a positive effect on residents’ altruistic behavior.

Bardsley [40] believed that responsibility was the influencing factor of the dictator’s altruistic behavior. Based on the bystander effect [41], Panchanathan, Frankenhuis, and Silk [42] designed the N-person dictator game by manipulating the number of dictators. That research showed that individuals began to become selfish with an increasing number of dictators. Besides, Cryder and Loewenstein [43] indirectly assessed the dictator’s level of responsibility by manipulating the clarity of the receiver’s information. The results showed that the clearer the information was, the more generous the dictator was. These studies revealed that as the information of the dictator and receiver was changed, the responsibility that the individual perceived was different, and the dictator’s sense of responsibility might have a positive correlation with his altruistic behavior. In comparison, the present study presented the identity information of receivers that enhanced the authenticity of receivers, and then clarified the responsibility of the dictator. Meanwhile, we directly measured participants’ responsibility by the SOC-RQ and validated the relationship between SOC-R and dictator giving. These results are consistent with previous research (such as Panchanathan et al. [42]). At the same time, the results also verified the validity of repeated DGs.

According to the theory of SOC-R [23], a person’s SOC-R is driven by their personal belief system. When faced with different community contexts, people will adjust their belief systems to meet the regulatory requirements of situations. At the same time, scholars believe that the sense of responsibility contains the dimension of morality itself [44,45]. In daily life, social norms reinforce the individual’s responsibility and morality, which in turn translates the helping intention into altruistic behavior [46]. As Fehr and Schmidt [33] found, social norms drove people to become altruistic in the DG. Besides, altruistic behavior is also related to the inhibition control in the executive function [47]. In the present study, a dictator with a high level of SOC-R may have stronger ability to suppress the motivation of maximizing personal interests. At the same time, the preference for equal income makes individuals selfish in repeated DGs [48]. Therefore, this study believes that social norms make individuals have fair beliefs and form the correct behavioral norms. Individuals with a higher SOC-R are clearer about their responsibility to receivers and have a higher fairness belief, and then become more altruistic in the DG.

In summary, repeated DGs are more scientific than a one-shot dictator game and can be used as an effective measurement paradigm for altruistic behavior. The higher the level of SOC-R is, the more money the individual offers to others (the higher the level of altruistic behavior). Findings from the present study can not only enrich theories of altruism in community psychology and economic psychology but also provide inspiration to community managers to enhancement residents’ altruistic behavior. First, community managers should help residents to develop correct social norms by moral education and community activities (such as helping others or blood donation). Second, community managers should enhance residents’ SOC-R from the dimensions of SOC-R; for example, helping residents to develop mutual help groups for the same problems to strengthen residents’ sense of community responsibility.

## 5. Conclusions

In conclusion, the present study showed that repeated DGs were an effective paradigm for measuring people’s altruistic behavior. As the number of DGs increased, the level of altruistic behavior of residents gradually decreased. Most importantly, SOC-R had a positive effect on people’s altruistic behavior. The present study proposed a new mechanism of altruistic behavior from the perspective of community psychology. It explained the altruistic behavior which could not be explained by kin altruism and reciprocal altruism. Meanwhile, the conclusions excluded the effect of participants’ empathy and the conflict of selfishness and social preferences with the help of the repeated DGs. Altruism is a natural trait of human beings [49]. In modern society, promoting altruistic behavior helps to build a more stable environment and reduce the occurrence of aggressive behavior. Moreover, research on SOC-R should be as important as research on family responsibility and social responsibility because SOC-R and altruistic behavior have a positive correlation with people’s well-being and physical and mental health [4,22,25,50]. Meanwhile, this study may also have important implications for the study of community environmental protection and community engagement.

However, several limitations should be noted. First, future studies should increase the number of receivers—for example, repeating the dictator game 12 times [16]—so that we can get a complete curve of average offers and calculate the rate of change as well as the turning point. Second, future research should improve the experimental design; on the one hand, to improve the authenticity of the experiment by changing the method of the presentation of receiver information—for example, presenting the videos of participants in another lab by a blurred image, which can not only ensure the anonymity of the experiment but also improve the authenticity of the experiment—on the other hand, to further validate the relationship between the SOC-R and altruistic behavior, longitudinal follow-up studies and cross-cultural studies should be considered. Researchers should record the participants’ number of altruistic behaviors in daily life after the repeated DGs. Besides, other influence factors should be considered in the future (such as personality traits). For example, Ben-Ner and Kramer [13] found that subjects who were extroverted and neurotic had a high level of altruism. Meanwhile, the relationship between SOC-R, self-control and altruistic behavior should be explored. Last but not least, future research can enrich the model of the impact mechanism of the SOC-R on altruistic behavior, explore possible mediating and moderating variables, and determine possible cognitive causes with the help of event-related potential technique and functional magnetic resonance imaging.

## Figures and Tables

**Figure 1 ijerph-17-00460-f001:**
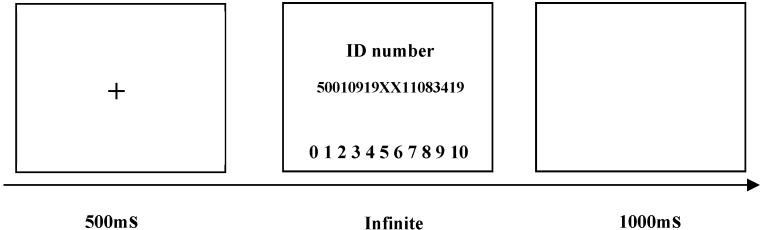
The experimental process of repeated dictator games.

**Figure 2 ijerph-17-00460-f002:**
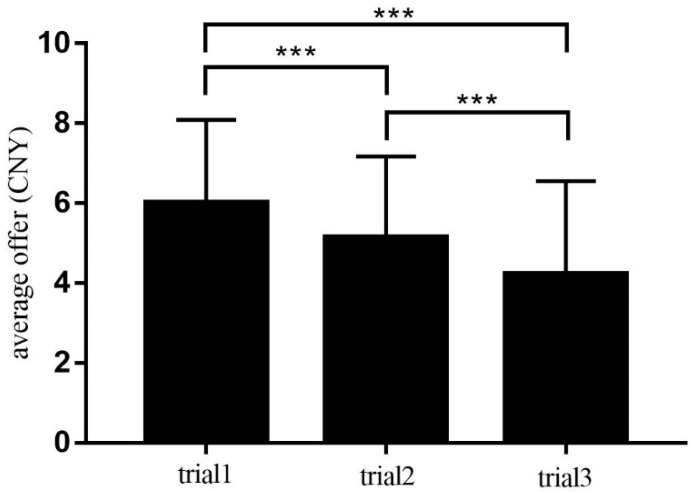
The differences in the average offer between the three trials. Note: *** *p* < 0.001.

**Figure 3 ijerph-17-00460-f003:**
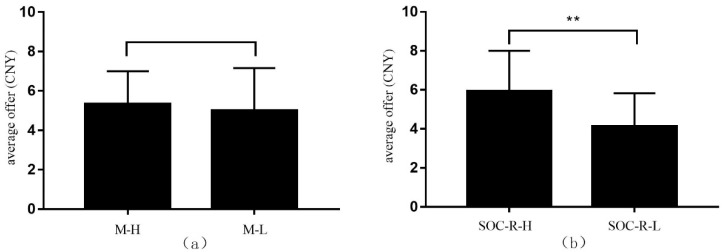
The differences in the average offer on money motivation and sense of community responsibility (SOC-R). (**a**) Test of difference in altruism level between high and low money motivation groups; (**b**) Test of difference in altruism level between high and low SOC-R groups. *Note:* M-H = group with high money motivation, M-L = group with low money motivation; SOC-R-H = group with a high level of sense of community responsibility, SOC-R-L = group with a low level of sense of community responsibility (** *p* < 0.01).

**Table 1 ijerph-17-00460-t001:** The results of the descriptive statistical analysis of dictators’ giving.

	*M*	*SD*	*Min*	*Max*
Trial 1	6.01	2.08	2	10
Trial 2	5.15	2.02	2	10
Trial 3	4.23	2.32	0	10
All trials	5.13	1.93	2.33	10

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
