# Peer review of "The Effect of Sense of Community Responsibility on Residents’ Altruistic Behavior: Evidence from the Dictator Game"

_ijerph, 2020, doi:10.3390/ijerph17020460_

Round 1

Reviewer 1 Report

The authors cite a decline in "people's concern for others" between 1979 and 2010.  2010 is a decade ago.  What do the results of current studies show?  There is a citation (2012) of civic values, including social issues and environmental protection behaviors, being lost in a survey of 8.7 million Americans.  It is not clear what "lost" means. Obviously, given the online behaviors of Americans today, both are strong. However, the research involves Chinese citizens, who come from a different culture than Americans, so it is not clear what the comparison is for.

The paper reads as an outline of a proposed study, and a summary of prior research.  The culture of the people that the survey was given to is important.  What was their behavior in the past?  There is no explanation of the dictator game, and how it was used.  The questions were not included in the study.  The two items in 2.2 Measures are not explained in terms of the study. The results are not explained.  Altruistic behavior is not fully defined, nor is community responsibility.

This is definitely not ready for publication in a journal.

Author Response

Dear reviewer,

Thank you so much for your advice on our paper (ijerph-665704). We are grateful to you for the detailed feedback which enabled us to enhance the manuscript. We have carefully addressed each of the comments below and highlighted (in red) the main changes made in the revised paper. Thank you for the opportunity to resubmit our paper for further consideration.

Response to Reviewer 1 Comments

Point 1: The authors cite a decline in "people's concern for others" between 1979 and 2010.  2010 is a decade ago.  What do the results of current studies show?  There is a citation (2012) of civic values, including social issues and environmental protection behaviors, being lost in a survey of 8.7 million Americans.  It is not clear what "lost" means. Obviously, given the online behaviors of Americans today, both are strong. However, the research involves Chinese citizens, who come from a different culture than Americans, so it is not clear what the comparison is for.

Response 1: We appreciated the reviewer raising these concerns. We have combed and adjusted the structure of the manuscript to make the presentation clearer and more logical. First, we have added a report (the 2013 China General Social Survey) to page 1, paragraph 1. Second, the expression is “declined” in the study of Twenge et al., (2012). We have changed the “lost” into “declined”. Third, the first paragraph aims to describe changes in citizenship values and highlight the importance of research on responsibility and altruistic behavior.

“Moreover, the 2013 China General Social Survey (CGSS) shown that only 32% of the 5,734 residents joined in donations voluntarily and unconditionally, and only 7% of the residents participated in social volunteer service activities (the survey could be accessed online via http://www.cnsda.org/index.php?r=projects/view&id=93281139).”

“Meanwhile, researchers found that people’s civic values (such as concern about social issues and environmental protection behaviors) also declined [2]”

Point 2: The paper reads as an outline of a proposed study, and a summary of prior research.  The culture of the people that the survey was given to is important.  What was their behavior in the past?  There is no explanation of the dictator game, and how it was used.  The questions were not included in the study.  The two items in 2.2 Measures are not explained in terms of the study. The results are not explained.  Altruistic behavior is not fully defined, nor is community responsibility.

Response 2: We appreciated the reviewer raising these concerns. First, we have added a description of altruistic behavior of Chinese residents basing on the 2013 China General Social Survey (CGSS) on page 1, paragraph 1. Second, we have added more introduction of DG on page 2, paragraph 2 and paragraph 3. Moreover, we have summarized existing research and highlighted research issues basing on the theory of SOC-R and social norms. However, we are sorry that we don’t understand what the two items in 2.2 Measures are not explained means. The SOC-RQ consists of 22 items across three dimensions. Although we only gave three examples of items, but SOC-R was assessed by the total score of SOC-RQ. Third, we have added the introduction of altruistic behavior and SOC-R on page 2.

“Previous studies have suggested that DG is an effective paradigm to evaluate an individual’s altruistic behavior [13,14]. It is an economic game in which two players unknown to each other. The standard paradigm is a one-shot task [12]. At the beginning of the game, the participants will be divided into dictators and receivers. Meanwhile, the dictator will receive a certain amount of money from the experimenter. And he can dispense any amount of money to the receiver. Whatever the amount is, the receiver can only accept and has no power to reject the proposal. The more money the dictator allocates to the receiver, the more altruistic he is. A meta-analysis finds that dictator usually allocates 28.35% of the amount to the receiver [15]. Comparing with the ultimatum game in which receiver can reject the proposal, DG eliminates the dictator’s fear of the receiver’s rejection and strategy processing based on reciprocity motivation [16].”

“Previous studies mainly explained the mechanism of altruistic behavior from an evolutionary-theoretical perspective. They thought there were two types of altruism: kin altruism [7] and reciprocal altruism [8]. This means people tend to help relatives and persons with whom they exchange benefits, to ensure that family genes are passed on to future generations and that their psychology is kept in balance between giving and receiving. Besides, they argued that empathy [9] and a sense of guilt [10] also affected individual’s altruistic behavior. The majority of research conducted on altruistic behavior used the Self-Report Altruism scale (SRA scale) [11] and dictator game (DG) [12]. The SRA scale is a unidimensional scale comprising 20 items describing altruistic acts in various scenarios, and responses are given on a 5-point Likert scale ranging from “never” to “always”. While some items of SRA scale are not suitable for community residents and some scenarios are not common in China. For example, “I have helped a classmate who I did not know very well with a homework assignment because my knowledge of the topic was greater than his/hers.” Therefore, we chose the DG.”

“Thus, researchers developed the repeated DGs in which dictator needed to complete multiple rounds of assignments with a different partner for each trial [16,19]. Meanwhile, previous research usually used college students as subjects [18]. While a meta-analysis found that non-student participants gave much more money to the receiver than student participants [15]. Therefore, the related conclusions of students may not apply to non-students (e.g., residents) and the repeated DGs is more suitable for assessing an individual’s altruistic behavior.”

“A wide range of research has been done on the SOC-R [20-22]. It refers to a feeling of selfless personal responsibility for both the individual and the collective well-being of a community [20,23]. It originates from scholars' discussion of the sense of community (SOC) [20,23]. It is connected but different from SOC. Specifically, it is dominated by personal information systems, which reflects the sense of responsibility of residents to the community and others. While SOC bases on the theory of needs, emphasizing the importance of the community to meet people’s physical and psychological needs. SOC-R makes the research on SOC more perfect and deeper [21]. They are all very important concepts in community psychology.”

We would like to thank the reviewer and editor again for their critical yet constructive comments. We believe their feedback has resulted in important revisions and an improved manuscript. For more details, please see the revised paper. Thank you for the opportunity to revise our manuscript for continued consideration for publication in International Journal of Environmental Research and Public Health.

Sincerely,

Reviewer 2 Report

Title: The Effect of Sense of Community Responsibility on 3 Residents’ Altruistic Behavior: Evidence from Dictator Game

Thank you for the opportunity to read and review this paper. The topic of the paper is relevant, but overall, I feel that the paper as currently written lacks sufficient theoretical focus and clarity, or relatedly a strong theoretical contribution. My following comments are intended to be constructive and hope they are helpful to the authors as they move forward with this project.

-Introduction: this paper explores the association between sense of community responsibility (SOC-R) and altruistic behavior by repeated dictator games. Therefore, in the introduction I was looking forward to a clear research gap which was never forthcoming. The authors fail to succinctly state the gap their current study will be addressing. The introduction lacks of a discussion what motivates the authors to conduct this research. Does any research gap exist in existent literature? The motivation of this study should be made stronger in the introduction.

The paper in its current form doesn't provide a meaningful contribution to the extant literature. The contribution to the literature is unspecific and should be improved significantly. What exactly is new about your research? The authors should clearly demonstrate the contribution of this paper and in what manner it is different from previous papers. The authors underestimate their work by not stating their contribution properly and they should make a better statement of their contribution at the introduction section. Additionally, what is the theoretical contribution of your paper?

-Literature Review: Where is your theoretical section.? I was expecting a theoretical background supporting your hypotheses. A theoretical foundation must be developed, focusing specifically on the core of your topic. The authors allude to previous empirical studies, but the hypotheses lack of a real theoretical support with a solid perspective or approach. There are not enough theoretical reasonings that explain your hypotheses.

-Methods: This section is well explained.

-Results: The discussions of the results should be enriched.

-Discussions: This section is well explained.

-Conclusions: This section should include the objective and main results presented in this study. I also have missed a paragraph focused on the academic, political, managerial and economic implications derived from your results. Moreover, this section should include limitations and future lines of research.

Good luck!!!

Author Response

Dear reviewer,

Thank you so much for your advice on our paper (ijerph-665704). We are grateful to you for the detailed feedback which enabled us to enhance the manuscript. We have carefully addressed each of the comments below, and highlighted (in red) the main changes made in the revised paper. Thank you for the opportunity to resubmit our paper for further consideration.

Response to Reviewer 2 Comments

Thank you for the opportunity to read and review this paper. The topic of the paper is relevant, but overall, I feel that the paper as currently written lacks sufficient theoretical focus and clarity, or relatedly a strong theoretical contribution. My following comments are intended to be constructive and hope they are helpful to the authors as they move forward with this project.

Author’s reply: We appreciate the reviewer’s positive evaluation of the manuscript.

Point 1: This paper explores the association between sense of community responsibility (SOC-R) and altruistic behavior by repeated dictator games. Therefore, in the introduction I was looking forward to a clear research gap which was never forthcoming. The authors fail to succinctly state the gap their current study will be addressing. The introduction lacks of a discussion what motivates the authors to conduct this research. Does any research gap exist in existent literature? The motivation of this study should be made stronger in the introduction.

Response 1: We appreciated the reviewer raising these concerns. We have combed and adjusted the structure of the manuscript to make the presentation clearer and more logical. At the beginning of Introduction, we have described changes in citizenship values and highlighted the importance of research on responsibility and altruistic behaviour on page 1, paragraph 1. Meanwhile, we have summarized existing research and highlighted research issues basing on the theory of SOC-R and social norms on page 3.

“Many meta-analytic, horizontal and vertical studies shown that people's concern for others began to decrease since 1979 and reached the largest decline between 2000 and 2010 [1]. Meanwhile, researchers found that people’s civic values (such as concern about social issues and environmental protection behaviors) also declined [2]. Moreover, the 2013 China General Social Survey (CGSS) shown that only 32% of the 5,734 residents joined in donations voluntarily and unconditionally, and only 7% of the residents participated in social volunteer service activities (the survey could be accessed online via http://www.cnsda.org/index.php?r=projects/view&id=93281139). In summary, with the development of society, people’s citizenship cognition and behavior may be out of sync with the rapid growth of material consumption, especially people living in the developing countries. How to make individuals feel responsible and be caring for society and others should be a topic of common concern for current researchers and managers.”

“In the past, researchers paid little attention to the altruistic behavior and SOC-R of residents (especially Chinese residents). Despite rich literature on altruism, this is the first study exploring the association between SOC-R and altruistic behavior by repeated DGs. Based on the theory of SOC-R and social norms as well as the relationship between responsibility and altruism, the present study aimed to verify the relationship between SOC-R and altruistic behavior so as to provide theoretical reference for the cultivation of residents’ altruistic behavior.”

Point 2: The paper in its current form doesn't provide a meaningful contribution to the extant literature. The contribution to the literature is unspecific and should be improved significantly. What exactly is new about your research? The authors should clearly demonstrate the contribution of this paper and in what manner it is different from previous papers. The authors underestimate their work by not stating their contribution properly and they should make a better statement of their contribution at the introduction section. Additionally, what is the theoretical contribution of your paper?

Response 2: We appreciated the reviewer raising these concerns. We have added the theoretical contribution to the Conclusions on page 7, paragraph 3.

“In conclusion, the present study showed that the repeated DGs was an effective paradigm for measuring people’s altruistic behavior. As the round of DGs increased, the level of altruistic behavior of residents gradually decreased. Most importantly, SOC-R had a positive effect on people’s altruistic behavior. The present study proposed a new mechanism of altruistic behavior from the perspective of community psychology. It explained the altruistic behavior which could not be explained by kin altruism and reciprocal altruism. Meanwhile, the conclusions excluded the effect of participants’ empathy and the conflict of selfishness and social preferences with the help of the repeated DGs. Altruism is a natural trait of human beings [49]. In modern society, promoting altruistic behavior helps to build a more stable environment and reduce the occurrence of aggressive behavior. Moreover, research on SOC-R should be as important as research on family responsibility and social responsibility. Because SOC-R and altruistic behavior have positive correlation with people’s well-being and physical and mental health [4,22,25,50]. Meanwhile, this study may also have important implications for the study of community environmental protection and community engagement.”

Point 3: Where is your theoretical section.? I was expecting a theoretical background supporting your hypotheses. A theoretical foundation must be developed, focusing specifically on the core of your topic. The authors allude to previous empirical studies, but the hypotheses lack of a real theoretical support with a solid perspective or approach. There are not enough theoretical reasonings that explain your hypotheses.

Response 3: We appreciated the reviewer raising this concern. We have added the theoretical background to support our hypotheses on page 3, paragraph 2.

“At present, the direct relationship between SOC-R and altruistic behavior is not known. Hoffman [27] thought altruism was a sense of responsibility for others and the manifestation of the development of individual responsibility. Brebels, De Cremer, and Sedikides [28] deemed that the perception of responsibility was the basis of altruism and the activation of social norms. Moreover, research on children’s altruism showed that the responsible group expressed more altruism than the non-responsible group [29]. A study demonstrated that social responsibility was a determinant of altruistic behavior [30]. Individuals with stronger sense of family responsibility reported more altruistic behavior [31]. These studies indicate that there is a relationship between responsibility and altruistic behavior. According to the theory of SOC-R [23], personal belief system makes people’s behavior conform to social norms. The core of this theory is that social norms make people's cognition and behavior coordinate, and then adapt to normative requirements of different social situations. The early theory of social norms believed that individual’s altruistic behavior was influenced by people’s perception of responsibility of social norms [32]. Social norms make people altruistic in the DG [18,33]. Moreover, a study indicated that sense of responsibility had cross-context consistency, an individual’s sense of responsibility in one context (e.g., family) had strong predictive effect on his sense of responsibility in another context [34]. People not only have a need to be a part of community but also feel a sense of duty and responsibility toward their community [35,36]. Therefore, perhaps it can speculate on the relationship between SOC-R and altruistic behavior from studies of family responsibility and social responsibility.”

Point 4: This section (Conclusion) should include the objective and main results presented in this study. I also have missed a paragraph focused on the academic, political, managerial and economic implications derived from your results. Moreover, this section should include limitations and future lines of research.

Response 4: We appreciated these suggestions. We have added the implications and limitations and future directions to Conclusions on page 7, paragraph 3 and paragraph 4.

“In conclusion, the present study showed that the repeated DGs was an effective paradigm for measuring people’s altruistic behavior. As the round of DGs increased, the level of altruistic behavior of residents gradually decreased. Most importantly, SOC-R had a positive effect on people’s altruistic behavior. The present study proposed a new mechanism of altruistic behavior from the perspective of community psychology. It explained the altruistic behavior which could not be explained by kin altruism and reciprocal altruism. Meanwhile, the conclusions excluded the effect of participants’ empathy and the conflict of selfishness and social preferences with the help of the repeated DGs. Altruism is a natural trait of human beings [49]. In modern society, promoting altruistic behavior helps to build a more stable environment and reduce the occurrence of aggressive behavior. Moreover, research on SOC-R should be as important as research on family responsibility and social responsibility. Because SOC-R and altruistic behavior have positive correlation with people’s well-being and physical and mental health [4,22,25,50]. Meanwhile, this study may also have important implications for the study of community environmental protection and community engagement.

However, several limitations should be noted. First, future studies should increase the number of receivers. For example, repeating the dictator game twelve times [16]. So that we can get the complete curve of average offers and calculate the rate of change as well as the turning point. Second, future research should improve the experimental design. On the one side, to improve the authenticity of the experiment by changing the way of the presentation of receivers’ information. For example, presenting the videos of participants in another lab by blurred image. This way can not only ensure the anonymity of the experiment but also improve the authenticity of experiment. On the other side, to further validate the relationship between the SOC-R and altruistic behavior, longitudinal follow-up studies and cross-cultural studies should be considered. Researchers should record participants’ number of altruistic behaviors in daily life after the repeated DGs. Besides, other influence factors should be considered in the future (such as personality traits). For example, Ben-Ner and Kramer [13] found that subjects who were extraversion and neuroticism had a high level of altruism. Meanwhile, the relationship between SOC-R, self-control and altruistic behavior should be explored. Last but not least, future research can enrich the model of the impact mechanism of the SOC-R on altruistic behavior, explore possible mediating and moderating variables, and find out possible cognitive causes with the help of ERP and fMRI techniques.”

We would like to thank the reviewer and editor again for their critical yet constructive comments. We believe their feedback has resulted in important revisions and an improved manuscript. For more details, please see the revised paper. Thank you for the opportunity to revise our manuscript for continued consideration for publication in International Journal of Environmental Research and Public Health.

Sincerely,

Reviewer 3 Report

This is a very good paper about a highly relevant topic. Indeed, the sustainability translated into CSR is critical for our global society and the Authors are to be complimented to go into deeper understanding and testing the relationship between SCR and altruism. Intuitively, it is often assumed that individuals more inclined to accept responsibility and as well more advanced vis-a-vis the concept of fairness and altruism and ultimately their responsibility should lead to more justice (both arithemetic and geometric). However, there are few studies addressing in a robust this intuitive expectation. Hence, the assessment of the SCR based on questionnaires and altruism based on DG via digital methods and their comparison leading to a co-relation is an excellent idea. Naturally, the paper could be (and perhaps should) be further improved to make this contribution truly valuable and in this respect I have the following recommendations:

please explain more the dictator game (DG) and why this is a good tool for the altruism assessment please explain more why repeated DGs provide a more complete picture please explain why did you select exactly this sample of 95 residents and provide more information about them please expand more your 4.Discussion and 5.Conclusion (especially the 5.Conclusion is TOO short!!!!), because your results allow it and it would be pity to skip it please consider whether your results are globally relevant or rather linked to Southwest China discuss any deviations or surprises (in)directly implied by your study use more up-to-date references

Author Response

Dear reviewer,

Thank you so much for your advice on our paper (ijerph-665704). We are grateful to you for the detailed feedback which enabled us to enhance the manuscript. We have carefully addressed each of the comments below, and highlighted (in red) the main changes made in the revised paper. Thank you for the opportunity to resubmit our paper for further consideration.

Response to Reviewer 3 Comments

This is a very good paper about a highly relevant topic. Indeed, the sustainability translated into CSR is critical for our global society and the Authors are to be complimented to go into deeper understanding and testing the relationship between SCR and altruism. Intuitively, it is often assumed that individuals more inclined to accept responsibility and as well more advanced vis-a-vis the concept of fairness and altruism and ultimately their responsibility should lead to more justice (both arithemetic and geometric). However, there are few studies addressing in a robust this intuitive expectation. Hence, the assessment of the SCR based on questionnaires and altruism based on DG via digital methods and their comparison leading to a co-relation is an excellent idea. Naturally, the paper could be (and perhaps should) be further improved to make this contribution truly valuable and in this respect I have the following recommendations:

Author’s reply: We appreciate the reviewer’s positive evaluation of the manuscript.

Point 1: please explain more the dictator game (DG) and why this is a good tool for the altruism assessment please explain more why repeated DGs provide a more complete picture please explain why did you select exactly this sample of 95 residents and provide more information about them please expand more your 4.Discussion and 5.Conclusion (especially the 5.Conclusion is TOO short!!!!), because your results allow it and it would be pity to skip it please consider whether your results are globally relevant or rather linked to Southwest China discuss any deviations or surprises (in)directly implied by your study use more up-to-date references

Response 1: We appreciated these suggestions. First, we added more introduction of DG and explained the validity of the repeated DGs on page 2. Second, recruitment of participants is mainly through the convenience sampling method. We didn't preset the sample size at first. All participants volunteered to participate in the experiment after seeing the community poster. And we have added some information about the subjects. Third, we have enriched the Conclusions on page 7, paragraph 3 and paragraph 4.

“Previous studies have suggested that DG is an effective paradigm to evaluate an individual’s altruistic behavior [13,14]. It is an economic game in which two players unknown to each other. The standard paradigm is a one-shot task [12]. At the beginning of the game, the participants will be divided into dictators and receivers. Meanwhile, the dictator will receive a certain amount of money from the experimenter. And he can dispense any amount of money to the receiver. Whatever the amount is, the receiver can only accept and has no power to reject the proposal. The more money the dictator allocates to the receiver, the more altruistic he is. A meta-analysis finds that dictator usually allocates 28.35% of the amount to the receiver [15]. Comparing with the ultimatum game in which receiver can reject the proposal, DG eliminates the dictator’s fear of the receiver’s rejection and strategy processing based on reciprocity motivation [16]. Besides, comparing with real donor behavior, DG can more accurately reflect the individual’s altruistic behavior because of its anonymity and its direct involvement with the individual’s interests. While real donor behavior is susceptible to the level of individual empathy [17].

However, researchers found that the level of individual altruism changed dynamically. Scholars found that the dictator allocated less money to the receiver when there were more receivers [18], which meant people would become selfish in the repeated dictator games (DGs). Researchers thought it was not possible to determine one’s level of altruism just relying on the amount of money he/she gave in the one-shot DG [16]. Thus, researchers developed the repeated DGs in which dictator needed to complete multiple rounds of assignments with a different partner for each trial [16,19]. Meanwhile, previous research usually used college students as subjects [18]. While a meta-analysis found that non-student participants gave much more money to the receiver than student participants [15]. Therefore, the related conclusions of students may not apply to non-students (e.g., residents) and the repeated DGs is more suitable for assessing an individual’s altruistic behavior.”

“Participants (N = 95, 30% male) were urban residents from Southwest China. They came from many different communities and were recruited through community posters. Participants ranged between 18 and 50 years old (M = 33.20, SD = 9.22). Most of them were Han ethnicity (95.79%), reported no religious beliefs (88.42%), owned their homes (92.63%), and lived in current communities for more than 10 years (66.30%). Their vision or corrected vision was normal. And they had no color blindness, weak color, and were familiar with the basic operation of the computer.”

“In conclusion, the present study showed that the repeated DGs was an effective paradigm for measuring people’s altruistic behavior. As the round of DGs increased, the level of altruistic behavior of residents gradually decreased. Most importantly, SOC-R had a positive effect on people’s altruistic behavior. The present study proposed a new mechanism of altruistic behavior from the perspective of community psychology. It explained the altruistic behavior which could not be explained by kin altruism and reciprocal altruism. Meanwhile, the conclusions excluded the effect of participants’ empathy and the conflict of selfishness and social preferences with the help of the repeated DGs. Altruism is a natural trait of human beings [49]. In modern society, promoting altruistic behavior helps to build a more stable environment and reduce the occurrence of aggressive behavior. Moreover, research on SOC-R should be as important as research on family responsibility and social responsibility. Because SOC-R and altruistic behavior have positive correlation with people’s well-being and physical and mental health [4,22,25,50]. Meanwhile, this study may also have important implications for the study of community environmental protection and community engagement.

However, several limitations should be noted. First, future studies should increase the number of receivers. For example, repeating the dictator game twelve times [16]. So that we can get the complete curve of average offers and calculate the rate of change as well as the turning point. Second, future research should improve the experimental design. On the one side, to improve the authenticity of the experiment by changing the way of the presentation of receivers’ information. For example, presenting the videos of participants in another lab by blurred image. This way can not only ensure the anonymity of the experiment but also improve the authenticity of experiment. On the other side, to further validate the relationship between the SOC-R and altruistic behavior, longitudinal follow-up studies and cross-cultural studies should be considered. Researchers should record participants’ number of altruistic behaviors in daily life after the repeated DGs. Besides, other influence factors should be considered in the future (such as personality traits). For example, Ben-Ner and Kramer [13] found that subjects who were extraversion and neuroticism had a high level of altruism. Meanwhile, the relationship between SOC-R, self-control and altruistic behavior should be explored. Last but not least, future research can enrich the model of the impact mechanism of the SOC-R on altruistic behavior, explore possible mediating and moderating variables, and find out possible cognitive causes with the help of ERP and fMRI techniques.”

We would like to thank the reviewer and editor again for their critical yet constructive comments. We believe their feedback has resulted in important revisions and an improved manuscript. For more details, please see the revised paper. Thank you for the opportunity to revise our manuscript for continued consideration for publication in International Journal of Environmental Research and Public Health.

Sincerely,

Round 2

Reviewer 1 Report

There are many incomplete sentences that begin with "While." They must be corrected.

Thank you for making the suggested revisions.  The added information will make the study much more valuable to readers.

Given the Chinese government's support for altruistic behavior, this may prove to be an important study.

Author Response

Dear reviewer,

Thank you so much for your good suggestions on our paper (ijerph-665704). We are grateful to you for the detailed feedback which enabled us to enhance the manuscript. We have carefully addressed each of the comments below and highlighted (in red) the main changes made in the revised paper. Thank you for the opportunity to resubmit our paper for further consideration.

Response to Reviewer 1 Comments

Point 1: There are many incomplete sentences that begin with "While." They must be corrected.

Response 1: We appreciated the reviewer raising this concern. We have transformed “While” into “However”, “Yet”, and “On the contrary”.

Point 2: Thank you for making the suggested revisions. The added information will make the study much more valuable to readers. Given the Chinese government's support for altruistic behavior, this may prove to be an important study.

Response 2: We appreciated this good suggestion. We have added the Chinese government’s support for altruistic behavior on page 1.

“In 2017, the Chinese government proposed the abandonment of egoism and believed that altruism is an effective way to build a community of human destiny.”

We would like to thank the reviewer and editor again for their critical yet constructive comments. We believe their feedback has resulted in important revisions and an improved manuscript. For more details, please see the revised paper. Thank you for the opportunity to revise our manuscript for continued consideration for publication in International Journal of Environmental Research and Public Health.

Sincerely,

Reviewer 2 Report

I appreciate the opportunity to read the paper entitled “The Effect of Sense of Community Responsibility on Residents’ Altruistic Behavior: Evidence from
Dictator Game ”.

I have read the paper and I think it is interesting and has potential to publish in the International Journal Environmental Responsibility Public Health. Authors have improved all the aspects commented previously. For this reason, I think this manuscript should be accept in this form.

Good luck with your revision.

Author Response

Dear reviewer,

Thank you so much for your positive evaluation of our paper (ijerph-665704). We are grateful to you for your hard work and decision.

Response to Reviewer 2 Comments

I appreciate the opportunity to read the paper entitled “The Effect of Sense of Community Responsibility on Residents’ Altruistic Behavior: Evidence from Dictator Game”.

I have read the paper and I think it is interesting and has potential to publish in the International Journal of Environmental Research and Public Health. Authors have improved all the aspects commented previously. For this reason, I think this manuscript should be accept in this form.

Good luck with your revision.

Author’s reply: We appreciate the reviewer’s positive evaluation of the manuscript.

We would like to thank the reviewer and editor again for their critical yet constructive comments. We believe their feedback has resulted in important revisions and an improved manuscript. For more details, please see the revised paper. Thank you for the opportunity to revise our manuscript for continued consideration for publication in International Journal of Environmental Research and Public Health.

Sincerely,
